# Long-Term High-Fat Diet Consumption Induces Cognitive Decline Accompanied by Tau Hyper-Phosphorylation and Microglial Activation in Aging

**DOI:** 10.3390/nu15010250

**Published:** 2023-01-03

**Authors:** Zheng Liang, Xiaokang Gong, Runjia Ye, Yang Zhao, Jin Yu, Yanna Zhao, Jian Bao

**Affiliations:** 1Institutes of Biomedical Sciences, School of Medicine, Jianghan University, Wuhan 430056, China; 2Department of Pathology and Pathophysiology, School of Medicine, Jianghan University, Wuhan 430056, China

**Keywords:** high-fat diet, cognitive decline, Tau hyperphosphorylation, microglial activation

## Abstract

High-fat diet (HFD) intake is commonly related to a substantial risk of cognitive impairment for senior citizens over 65 years of age, which constitutes a profound global health burden with several economic and social consequences. It is critical to investigate the effects of long-term HFD consumption on cognitive function and to inspect the potential underlying mechanisms. In the present study, 9-month-old male C57BL/6 mice were randomly assigned to either a normal diet (ND, 10 kcal% fat) or an HFD diet (60 kcal% fat) for 10 months. Then a series of behavioral tests, and histological and biochemistry examinations of the hippocampus and cortex proceeded. We found that long-term HFD-fed aged mice exhibited cognitive function decline in the object place recognition test (OPR). Compared with the ND group, the HFD-fed mice showed Tau hyperphosphorylation at ps214 in the hippocampus and at ps422 and ps396 in the cortex, which was accompanied by GSK-3β activation. The higher activated phenotype of microglia in the brain of the HFD group was typically evidenced by an increased average area of the cell body and reduced complexity of microglial processes. Immunoblotting showed that long-term HFD intake augmented the levels of inflammatory cytokines IL-6 in the hippocampus. These findings indicate that long-term HFD intake deteriorates cognitive dysfunctions, accompanied by Tau hyperphosphorylation, microglial activation, and inflammatory cytokine expression, and that the modifiable lifestyle factor contributes to the cognitive decline of senior citizens.

## 1. Introduction

Dietary intake of fat has markedly increased for the past few years and the global prevalence of obesity has doubled from 1980 to 2008 [1,2]. Numerous evidence highlights the harmful impact of diets rich in saturated fat and obesity that are associated with multiple diseases, such as type 2 diabetes (T2D), hyperlipidemia, hypertension, cardiovascular diseases, Alzheimer’s disease (AD), and psychiatric disorders [3,4,5]. Epidemiological studies have also indicated a tremendous risk for the development of mild cognitive impairment and dementia for those elders over 65 years of age who consume a long-term sustained and high intake of saturated fats over a 4- to 6-year period [6]. However, the mechanism of the impact of high-fat diet (HFD) consumption on the aging process has hitherto not been sufficiently elucidated.

Tau hyper-phosphorylation and aggregation are widely considered pathological events in plentiful neurodegenerative diseases, including Alzheimer’s disease (AD), frontotemporal dementia, and progressive supranuclear palsy [7]. Tau is a microtubule-associated protein that mainly localizes in the axonal compartments of neurons and has a critical role in stabilizing neuronal microtubules, thus regulating microtubule dynamics and promoting neurite outgrowth, and facilitating axonal transport under normal conditions [8]. However, while hyper-phosphorylation is under some pathological conditions, this abnormal hyper-phosphorylation tau disassociates from microtubules and aggregates into neurofibrillary tangles (NFTs), leading to dystrophic neurites [9]. Thus, the accumulation of hyper-phosphorylation tau is involved in synaptic neurotoxicity, neurodegeneration, and cognitive impairments [10,11].

Microglia are innate immune cells within the brain and play an important role in maintaining neural function, including surveying and optimizing the surrounding microenvironment, and maintaining homeostasis for neuronal survival and function [12,13,14]. Microglia promotes an optimal neural network under physiological conditions via synaptic remodeling, neurogenesis promotion, and cellular debris elimination [12,15,16,17,18]. However, microglia dysfunction is closely associated with inflammation and neuronal impairment, which is implicated in cognitive deficiency and brain-associated energy imbalance [19,20,21,22]. In the present study, we found that long-term HFD-feeding aged mice exhibited cognitive function decline compared to the ND (normal diet) feeding aged mice. In addition, long-term HFD induces Tau hyperphosphorylation, alters microglial morphology, and promotes inflammation in aged mice.

## 2. Materials and Methods

### 2.1. Animals

In the present study, nine-month-old male C57BL/6 mice (body weight 30–40 g, Vital River Laboratory Animal Technology Co., Ltd., Beijing, China) were randomly assigned into either a normal diet (ND) group, fed a standard diet (10 kcal% fat) or a high-fat diet (HFD) group, fed a high-fat diet (60 kcal% fat, D12492). All mice were housed in a temperature-controlled room at 22 ± 2 °C with a 12 h light/dark cycle (lights on at 8:00, light off at 20:00). After ten months of HFD or ND feeding, mice in the two groups were subjected to further ethological, morphological and biochemical analyzes. Mice from two groups fasted for 12 h, and then the blood glucose of the tail vein was measured by blood glucose test strip (Johnson & Johnson, New Brunswick, NJ, USA). The body weight of each group of mice was determined before the mice were sacrificed. The experiments were performed in conformity with the National Institutes of Health guide for the care and use of Laboratory animals (NIH Publications No. 8023, revised 1978), and the animal study protocol was approved by the Ethics Committee of Jianghan University (approval JHDXLL2022-072) for studies involving animals.

### 2.2. Object-Place Recognition

The Object-place recognition test was implemented as the previous study described [23]. Briefly, all mice were habituated in the action observing box (50 cm deep × 50 cm high × 50 cm wide) for five min in three consecutive daily sessions before the test. Each test session consisted of a probe phase and a test phase. In the probe phase, the animal was placed in the box for 5 min to explore two identical objects, which were placed on different corners. To eliminate smell interference, the box and objects were cleaned with 75% ethanol during the sessions. In the test phase, the animal was put into the box for another 5 min of free exploration. All the sessions were conducted by a technician who is blind to the groups. All data was recorded by the computerized video track system (XR-XJ117, XinRuan Information Technology Co. Ltd., Shanghai, China). For calculating the discrimination ratio, the time in exploration of the novel object minus the time in exploration of the familiar object, then divide the total object exploration time.

### 2.3. Fear Conditioning

The fear conditioning test was performed by a fear conditioning chamber with an aversive stimulus. The details of parameters for mouse training are as follows: 70 dB, 2 kHz tone duration is 30 s; the 0.6 mA shock duration is 2 s. On the training day (day 1), each mouse was sent in the fear conditioning chamber for free exploration 2 min before the delivery of a 30 s tone and light, which ended with a 2 s foot shock. The conditioned stimulus- unconditioned stimulus (CS-US) pair was delivered again with an interval of 2 min. On the second day for testing, each mouse was put into a fear conditioning chamber again with no CS or US for 3 min and the freezing time was recorded by the computerized video track system (XinRuan Information Technology Co. Ltd., Shanghai, China).

### 2.4. Barnes Maze

Barnes maze test was performed on a platform (91 cm diameter, elevated about 100 cm from the floor) consisting of 20 holes (each 5 cm in diameter). As the recessed escape box, a wooden plastic escape box (11 cm × 6 cm × 5 cm) was positioned beneath one of the holes. To motivate the mice to find the escape box, bright light, and four spatial cues were arranged around the platform. Each test session consisted of a probe phase and a test phase. For the probe phase (day 1–day 4), each mouse received two trials per day for the inter-trial interval (20–30 min). Each mouse was placed into a start tube located in the center of the maze to enter the escape box within 180 s in each trial. To consolidate the memories of the mice, they stayed in the escape box for 30 s. In the probe test (day 5), The target hole does not change its position, but the escape box was withdrawn from the platform. Each mouse was placed into a start tube and allowed to explore for 60 s. The error times, time to target the hole, and time spent in the target quadrant were recorded using the SuperMaze software (XinRuan Information Technology Co. Ltd., Shanghai, China).

### 2.5. Immunoblotting

After being euthanized, the cortical and hippocampal tissues of mice were rapidly removed. The total proteins from the hippocampus and cortex were extracted by using RIPA lysis buffer (Beyotime, Nantong, China) added phosphatase inhibitors, and protease inhibitors (Beyotime, China). Proteins were separated via sodium dodecyl sulfate-polyacrylamide gel electrophoresis and immediately transferred to nitrocellulose (NC) membranes (Millipore, Burlington, MA, USA). Then the proteins on NC membranes were blocked by the use of 5% nonfat milk for 1 h at room temperature. After protein blocking, the NC membranes were incubated with corresponding primary antibodies overnight (see Table 1). On the second day, the NC membranes were washed three times by TBST and then incubated with corresponding secondary antibodies for 1 h at room temperature. For detecting the protein expression, immune bands from NC membranes were taken a photo by an enhanced chemiluminescence system (Bio-Rad, Hercules, CA, USA).

### 2.6. Immunofluorescence

Briefly, the fixative brains were cut into sections (30 μm) by a vibratome stage (Leica VT1000S, Wetzlar, Germany). Brain slices were cleaned with phosphate-buffered solution (PBS, Beyotime, China) three times. Then slices were blocked with 0.1% Triton X-100 (Solarbio, Beijing, China) including 5% bovine serum albumin for 1 h at room temperature. Brain sections were incubated with corresponding primary antibodies overnight (see Table 1), and with appropriate secondary antibodies at 37 °C for 1 h. Then, sections were washed by TBST 5 times. Then the cell nucleus was stained by Hoechst (Invitrogen, Waltham, MA, USA) for 10 min and washed 3 times with PBS. In the end, the slices were sealed with an anti-fluorescence quencher (Beyotime, China). To acquire images, a laser scanning confocal microscope was used to take photos (Leica SP8, Germany). 3D reconstruction and statistics of microglia were quantified by Imaris 9.0 (Bitplane, Belfast, UK). To determine the branch tree morphology of microglia, Sholl analysis was used by placing concentric circles in 1 mm increments starting at 1 mm from the soma.

### 2.7. Fluoro-Jade C (FJC) Staining

FJC was used to detect degenerated neurons other than healthy neurons. Brain sections (40 μm) were loaded in gelatin-coated slides and dried on a slide warmer at 50–60 °C for 30 min. After fixed, sections were successively incubated in 1% sodium hydroxide for 5 min, then immersed in 70% ethanol for 2 min and washed with distilled water for 2 min. The sections were then immersed in 0.06% potassium permanganate for 10 min and distilled water for 2 min, followed by incubation in the 0.0004% Fluoro-Jade C labeling solution (Cat No. TR100-FJT, Amyjet Scientific Incorporation, Wuhan, China) for 10 min in the dark and washed by distilled water. Finally, the slides were placed on a slide warmer at 50–60 °C until they were completely dry. The dry slides were cleared with xylene and then covered with DPX (Cat No. M100). A fluorescence microscope (Olympus, Tokyo, Japan) was used to capture images.

### 2.8. Statistics

In this study, all data are represented as mean ± SEM. All statistical analyzes were done using GraphPad Prism8 (GraphPad Software Inc., Boston, MA, USA). For the significance test, two-tailed unpaired Student’s *t*-tests were employed for comparisons between the two groups. One-way ANOVA with repeated measures was performed with Dunn’s post hoc test when appropriate. Statistical significance was set at *p* ≤ 0.05.

## 3. Results

### 3.1. Long-Term HFD Deteriorates Behavioral Performance of Aged Mice

9-month-old C57BL/6J mice were fed an HFD or ND for 10 months. As expected after 10-month feeding, The body weight of HFD-fed mice was significantly increased compared to the matched ND group (Appendix A). Fasting blood glucose was also significantly increased in the HFD-fed group (Appendix A). Next, we conducted systematic behavioral analyzes to examine whether long-term HFD had any impact on behavioral indices. Object place recognition (OPR), fear conditioning, and Barnes maze to test hippocampus-dependent spatial learning and memory. In the test phase of the OPR task, HFD-feeding animals showed a significantly reduced exploration of the displaced object, suggesting that HFD feeding impaired the retention of spatial memory in the aged mice (Figure 1A,B). In the cohort for fear conditioning, the baseline of freezing was similar between the two groups, but the cued-induced freezing after 24 h observed a non-significant reduction in the HFD-feeding mice compared to the ND (Figure 1C,D). During the spatial learning test for Barnes Maze, no difference was observed in the primary errors to locate the target (Figure 1E). In a probe test performed 24 h after the last training day, the HFD feeding group showed an increasing tendency of the primary errors, (Figure 1F). Overall, these data suggest that long-term HFD deteriorates the spatial cognition of aged mice.

### 3.2. Long-Term HFD Induces Tau Hyperphosphorylation in Aged Mice

Tau hyper-phosphorylation exerts tau pathologies, which are associated with neurotoxicity, neurodegeneration, and cognitive impairments. We analyzed the levels of tau phosphorylation in the hippocampus and cortex. Tau phosphorylation at Ser214 (ps214) was significantly increased in the hippocampus of the HFD-fed mice compared to the ND (Figure 2A,B). Increased levels of Ser396-phospho-tau (ps396) and Ser422-phospho-tau (ps422) were observed in the cortex of HFD-treated mice (Figure 2C,D). The levels of total tau were comparable between the two groups in both hippocampus and cortex. These results suggest that long-term HFD leads to Tau hyperphosphorylation in the brain. Consistent with the increased tau phosphorylation within HFD treatment, immunofluorescence with anti-Ser199-phospho-tau (ps199) further showed an increased trend of Tau phosphorylation in DG (Figure 3A,B) and CA1(Figure 3A,C). In addition, FJC staining showed that the degenerated neurons were increased in HFD-fed mice compared to the matched ND group (Figure 3D–F).

GSK-3β is one of the key tau kinases, which is associated with tauopathology. GSK-3β phosphorylation at Ser9 (negative regulation) was decreased under HFD treatment, indicating HFD-induced GSK-3β activation in both hippocampus (Figure 4A,B) and cortex (Figure 4C,D). Other regulators of tau phosphorylation, including cyclin-dependent kinase 5 (CDK5), p35/p25 mitogen-activated protein kinase, and protein phosphatase 2A (PP2A) did not show any changes after HFD treatment (Figure 4A–D). Altogether, long-term HFD treatment promotes Tau phosphorylation via GSK-3β activation, which might be associated with cognitive deficits.

### 3.3. Long-Term HFD Alters Microglial Morphology and Promotes Inflammation in Aged Mice

It has been well-established that microglial dysfunction is implicated in neuronal plasticity, brain senescence, and multiple neurodegenerative diseases [20,21]. Microglial morphology, such as the complexity and the length of the processes, and the volume of the cell body, is inextricably related to their functions and changeable in the physiological process. [24,25], the phenotypes of microglia in the cortex and hippocampus were analyzed by using the microglial-specific marker Iba1 (Figure 5A). Compared with ND-fed mice, microglia from the HFD-fed mice exhibited significantly increased average territory area of the cell body in the cortex (Figure 5B) and DG (Figure 5C) and CA1 (Figure 5D) of the hippocampus, suggesting the alteration of microglial status induced by long-term HFD treatment. Sholl analysis of microglia in CA1 showed that the complexity of microglial processes was significantly decreased in the HFD-fed mice, again obviously illustrating a detrimental effect of HFD on microglial morphology (Figure 5E,F).

Microglial activation is related to systemic inflammation and inflammatory response, which plays a critical pathophysiological role in aging and cognitive deficits, and psychiatric disorders [3,26,27,28,29,30]. Aging-associated inflammatory cytokines, such as IL-6 and TNFα, were found to disrupt circuits that are involved in cognitive decline or dementia [31,32]. Immunoblotting showed that long-term HFD consumption augmented the expression level of IL-6 in the hippocampus (Figure 6A,B), although no differences between the two groups in the cortex (Figure 6C,D). Taken together, these results suggested that long-term HFD intake alters microglial morphology and escalates the levels of inflammatory cytokines in the hippocampus. We also observed there were no differences in levels of IL-6 and TNFα in the cortex after long-term HFD feeding. It might result from structural dissociation and functional distinction between the hippocampus and cortex [33,34]. However, further investigations are warranted to explore the underlying mechanism of different effects of HFD on the hippocampus and cortex.

### 3.4. Long-Term High-Fat Diet Does Not Activate Astrogliosis in Aged Mice

Astrocytes exert numerous detrimental effects through a process referred to as reactive astrogliosis, which is a pathological hallmark of several neurodegenerative diseases [35,36]. We assessed astrogliosis in the hippocampal CA1 (Figure 7A), DG (Appendix A), and cortex regions (Appendix A) by immunolabeling for GFAP (glial fibrillary acid protein). Statistical results showed that GFAP + cell number was not changed after long-term HFD feeding compared with ND-fed mice (Figure 7B). The GFAP % area covered was also unaffected by HFD feeding in any of the regions (Figure 7C). Taken together, these data show that long-term HFD consumption does not induce more brain astrogliosis in aged mice.

## 4. Discussion

Aging is an inevitable and complex physiological phenomenon and biological processes, which might initiate numerous progressive physiological and pathological alterations [37]. Within the central nervous system (CNS), aging is the leading risk factor for several neurodegenerative diseases that are closely associated with abnormal behavior [38]. Accumulating evidence suggests that HFD and the consequent metabolic dysfunctions may lead to mild cognitive impairment, AD, vascular dementia, and age-related cognitive decline [39]. In the present study, we found that long-term HFD-fed aged mice exhibited cognitive function decline compared to the ND-fed mice. In addition, long-term HFD induces Tau hyperphosphorylation, alters microglial morphology, and promotes inflammation in aged mice.

Tau hyperphosphorylation is considered a critical pathological event with diverse neurodegeneration-like consequences and behavioral disorders. Tau hyperphosphorylation and abnormal Tau accumulation are commonly observed in an age-dependent manner. HFD consumption has been speculated to lead to increasing tau phosphorylation through modulation of activities of tau kinases. Walker et al. reported ten-month HFD treatment increased the level of tau phosphorylation in the hippocampus of AD model mice [40]. Gong et al. also found that four-month HFD feeding for AD model mice increased insulin resistance and β-amyloid accumulation which is another pathological hallmark of AD [41]. Accumulating studies indicate that obese elderly people exhibit an increase in tau phosphorylation of the hippocampus. Here, we observed that ten-month HFD feeding significantly increases the level of tau phosphorylation in the hippocampus and cortex of C57BL/6J mice. GSK3β is the first identified tau kinase, whose dysregulated activity results in Tau hyper-phosphorylation at numerous sites, Aβ accumulation, and synaptic impairment [42]. At present, long-term HFD treatment induces GSK-3β activation that may facilitate tau hyper-phosphorylation in the brain of mice.

Microglia are brain macrophages and play a critical role in maintaining homeostasis and regulating neural function [12,13]. Dysfunction of microglia is involved in brain senescence and neurodegenerative diseases [43]. Increasingly, literature has demonstrated that long-term HFD consumption results in insulin resistance, neuroinflammation, microglial activation, and oxidative stress in the brain, which contribute to cognitive impairment and psychiatric behavioral abnormalities [12]. Bocarsly et al. report that HFD feeding increases microglial processes in the prefrontal cortex, leading to diminish dendritic spine and reduce synaptic protein [44]. Increased microglial synaptic engulfment in the hippocampus and impaired cognitive capacity have been observed in HFD-feeding mice; blocking microglial phagocytosis or inhibiting microglial activation rescue the loss of dendritic spine and cognitive impairment [45,46]. Microglial morphology is dynamically changeable and inextricably related to their functions [24,25]. At present, we observed that HFD feeding results in an increased average volume of the cell body and a decrease in the complexity of microglial processes that illustrate microglial activation. On the other hand, chronic systemic inflammation, including an elevation of peripheral inflammation and neuro-inflammation, has been suggested as a vital pathology in long-term HFD treatment and obesity [47]. At present, the levels of IL-6 and TNFα in the hippocampus are augmented within long-term HFD consumption. We also observed there were no differences in the cortex after long-term HFD feeding. It might result from structural dissociation and functional distinction between the hippocampus and cortex [33,34]. It needs further investigations to explore the underlying mechanism.

Astrocytes are the most populous glial cells in the CNS, which are essential to numerous physiological processes, including neuroprotection, development of synaptic plasticity, and homeostasis maintenance. Under inflammation and CNS insults, reactive astrogliosis has been regarded as a pathological hallmark of neurodegenerative diseases. Recent literature has suggested that HFD consumption results in reactive astrogliosis and hypothalamic inflammation; blocking astrocytic inflammation reduces hypothalamic inflammation in HFD-fed mice. However, the findings from our research revealed no difference in cell number and area covered by astrocytes in HFD and ND-feeding mice. This might result from: (i) different ages of mice tested; (ii) different durations of diet feeding; (iii) different kinds of diet. Therefore, further investigations are warranted to evaluate the effects of HFD on astrogliosis. In conclusion, we revealed that long-term HFD induces Tau tauopathology, alters microglial morphology, and promotes inflammation, leading to cognitive impairments in aged mice.

## 5. Conclusions

At present, we revealed that long-term HFD induces Tau tauopathology, alters microglial morphology, and promotes inflammation, leading to cognitive impairments in aged mice.

## Figures and Tables

**Figure 1 nutrients-15-00250-f001:**
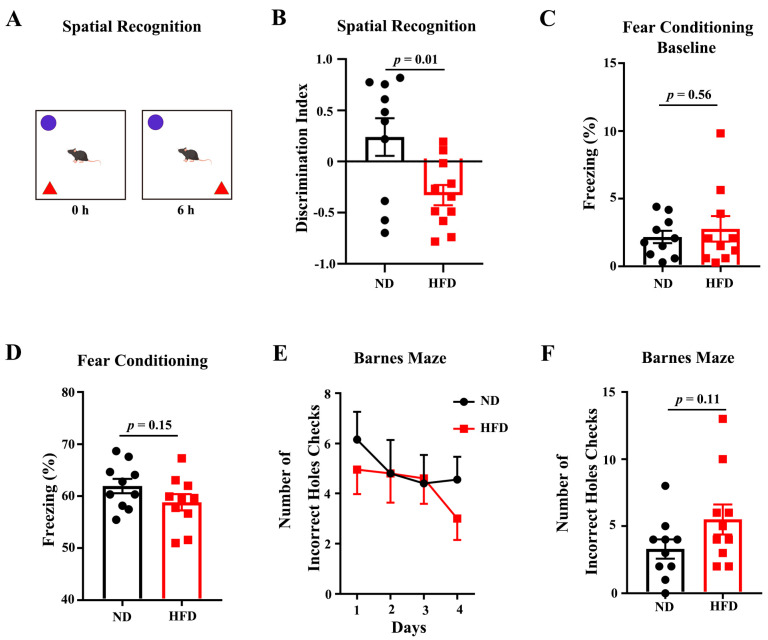
HFD feeding deteriorates the spatial recognition competence of aged mice. (**A**) Pattern diagram of OPR. (**B**) Discrimination index of normal diet (ND) mice and HFD feeding mice in OPR. (**C**,**D**) Baseline freezing rate and cue-induced freezing rate of ND-feeding mice and HFD-feeding mice. (**E**) Number of incorrect hole checks for two groups during training days of Barnes Maze. (**F**) Number of incorrect holes checked during the probe test. Data are presented as Mean ± SEM, (*n* = 10–11) for each group. The *p*-value for each set of data is annotated on the graph.

**Figure 2 nutrients-15-00250-f002:**
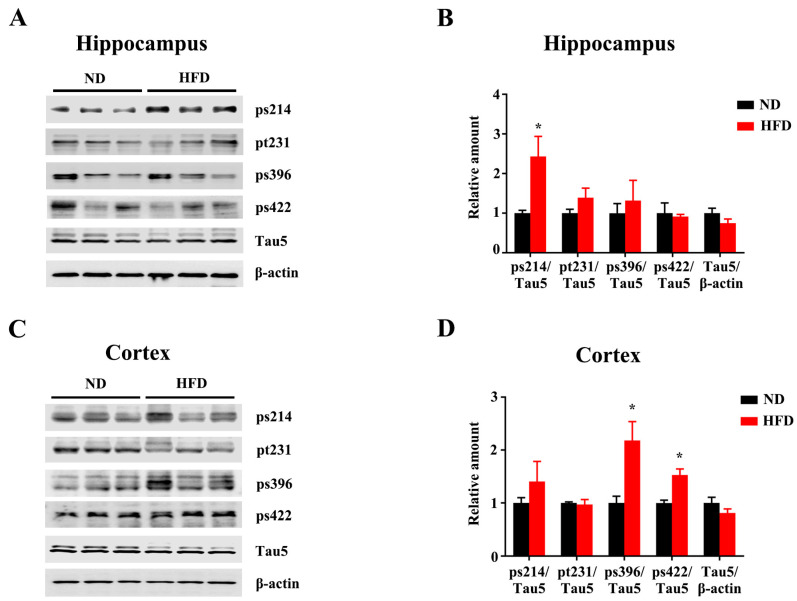
Long-term HFD feeding induces Tau hyperphosphorylation in aged mice. The expression of Tau phosphorylated at Ser214, Thr231, Ser396, Ser422, and total Tau from the hippocampus (**A**) and cortex (**C**) of ND and HFD feeding mice were evaluated by western blotting. (**B**,**D**) Quantitative statistics for (**A**,**C**) respectively. Data are presented as Mean ± SEM, (*n* = 3) for each group. * *p* < 0.05 vs. ND group.

**Figure 3 nutrients-15-00250-f003:**
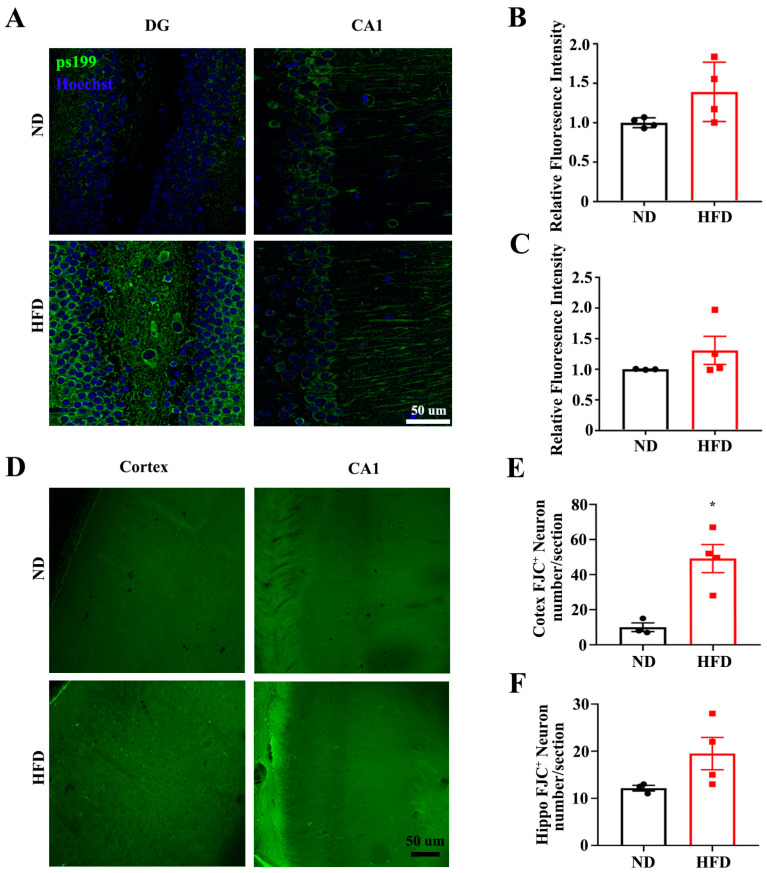
Long-term HFD feeding induces tauopathology in aged mice. Representative immunofluorescence images (**A**) and quantification of ps199 in CA1 (**A**), dentate gyrus (**B**), and CA3 (**C**) of ND or HFD-administrated mice. Representative images of FJC staining in Cortex and CA1 of ND or HFD administrated mice (scale bar = 50 μm). (**D**–**F**) FJC staining and quantitative statistics of positive cells in Cortex and CA1. Data are presented as Mean ± SEM, (*n* = 3/4) for each group. * *p* < 0.05 vs. ND group.

**Figure 4 nutrients-15-00250-f004:**
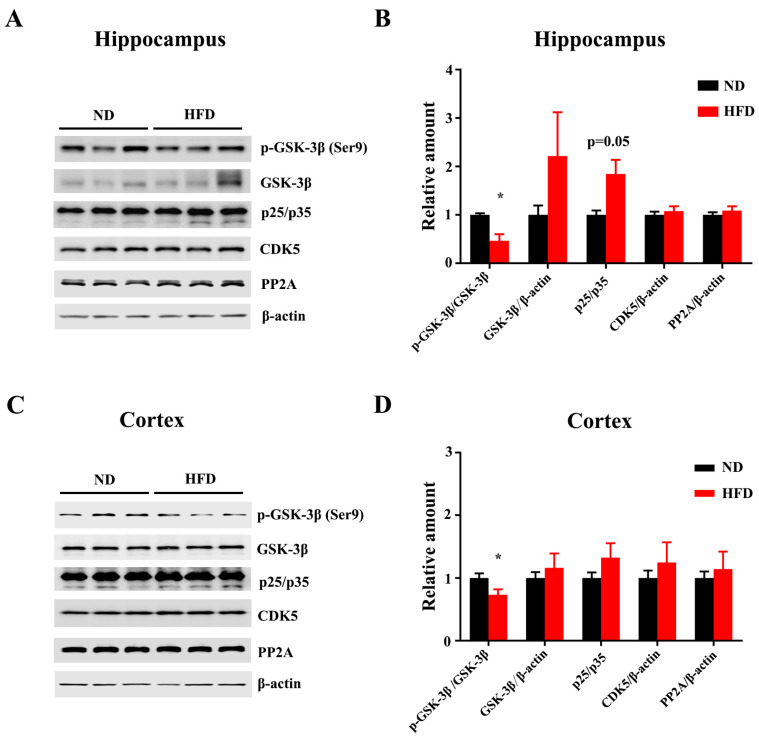
Long-term HFD feeding increases kinase activation of Tau in aged mice. The expression of GSK-3β phosphorylated at Ser9, total GSK-3β, p25, p35, CDK5, and PP2A from the hippocampus (**A**) and cortex (**C**) of ND and HFD feeding mice were evaluated by western blotting. (**B**,**D**) Quantitative statistics for (**A**,**C**) respectively. Data are presented as Mean ± SEM, (*n* = 3–6) for each group. * *p* < 0.05 vs. ND group.

**Figure 5 nutrients-15-00250-f005:**
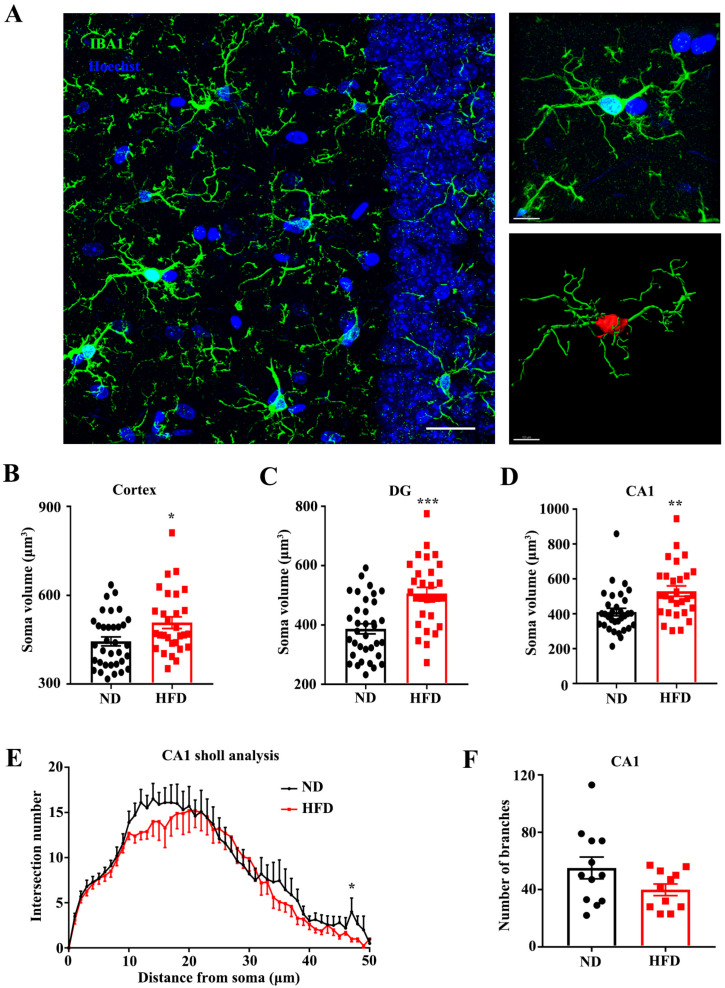
Long-term HFD feeding altered microglial morphology. (**A**) Left: Representative of confocal images for Iba1 + cells (scale bar = 30 μm). Right: The area and microglial processes were present (scale bar = 10 μm). (**B**–**D**) Quantitative statistics for soma volume of microglia in Cortex, DG and CA1 area. (**E**,**F**) The sholl analysis and the number of microglial branches in CA1. * *p* < 0.05, ** *p* < 0.01, *** *p* < 0.001 vs. ND group.

**Figure 6 nutrients-15-00250-f006:**
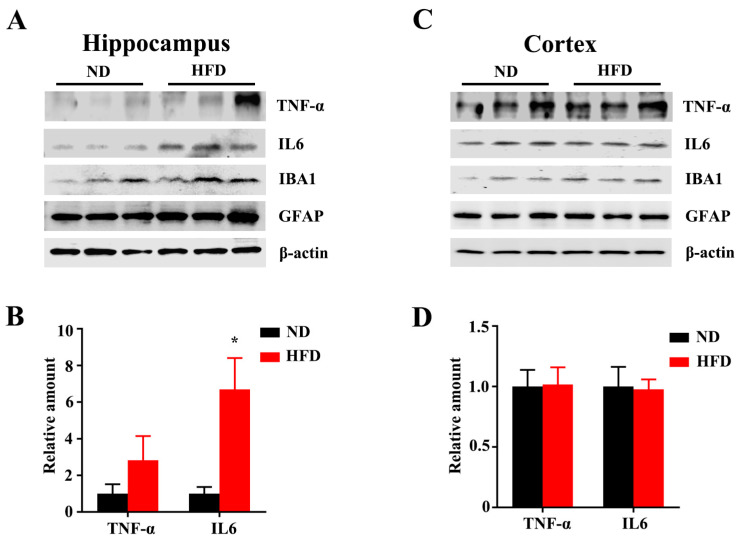
Long-term HFD feeding promotes inflammation in aged mice. The expression of TNF-α, IL6, IBA1, and GFAP from the hippocampus (**A**) and cortex (**C**) of ND and HFD-feeding mice was evaluated by western blotting. (**B**,**D**) Quantitative statistics for (**A**,**C**) respectively. Data are presented as Mean ± SEM, (*n* = 3) for each group. * *p* < 0.05 vs. ND group.

**Figure 7 nutrients-15-00250-f007:**
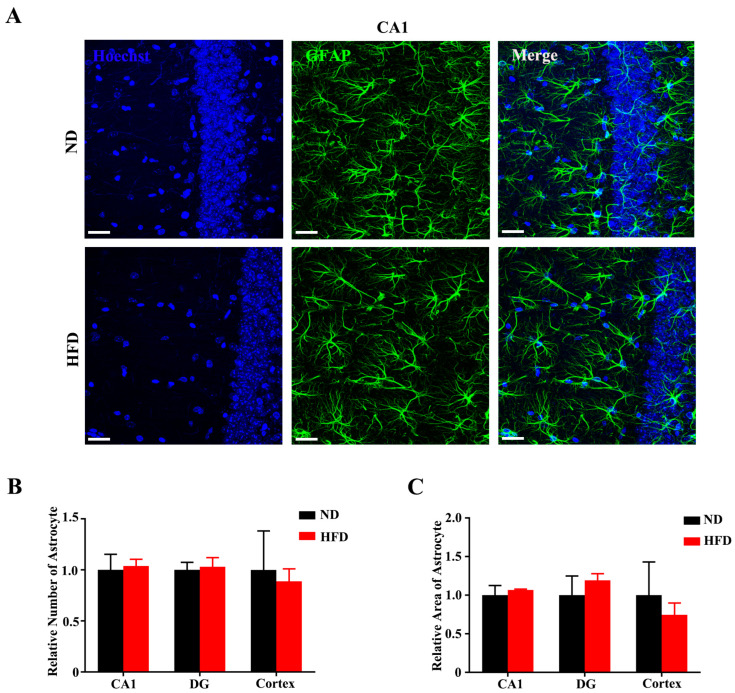
Long-term HFD feeding does not activate astrogliosis in aged mice. (**A**) The representative images of astrocytes in CA1 (scale bar = 30 μm). Quantitative statistics for number (**B**) and area (**C**) of GFAP+ cells in CA1, DG, and Cortex areas. Data are presented as Mean ± SEM, (*n* = 3) for each group.

**Table 1 nutrients-15-00250-t001:** The primary and secondary antibodies used in this study.

Antibody	Specificity	Type	Species	Source (Catalog Number)
Anti-tau (pS214)	Tau phosphorylated at Ser214	pAb	Rabbit	Immunoway (YP0259)
Anti-tau (pT231)	Tau phosphorylated at Thr231	pAb	Rabbit	Immunoway (YP0267)
Anti-tau (pS396)	Tau phosphorylated at Ser396	pAb	Rabbit	Immunoway (YP0263)
Anti-tau (pS422)	Tau phosphorylated at Ser422	pAb	Rabbit	Immunoway (YP0845)
Anti-tau (pS199)	Tau phosphorylated at Ser199	pAb	Rabbit	GeneTex (GTX24749)
Anti-Tau (Tau-5)	Total Tau	mAb	Mouse	Abcam (ab80579)
Anti-GSK3β (pS9)	GSK3β phosphorylated at Ser9	mAb	Rabbit	Cell Signaling (5558S)
Anti-GSK3β	Total GSK3β	mAb	Mouse	Immunoway (YM3633)
Anti-p25/p35	Total p25 and p35	mAb	Rabbit	Cell Signaling (2680S)
Anti-CDK5	Total CDK5	mAb	Rabbit	Cell Signaling (14145S)
Anti-PP2Ac	Total PP2A c subunit	mAb	Rabbit	Cell Signaling (2038S)
Anti-TNF-α	Total TNF-α	mAb	Mouse	Proteintech (60291-1-Ig)
Anti-IL-6	Total IL-6	pAb	Rabbit	Proteintech (21865-1-AP)
Anti-IBA1	Total IBA1	mAb	Rabbit	Abcam (ab178846)
Anti-GFAP(GA5)	Total GFAP	mAb	Mouse	Cell Signaling (3670S)
Anti-Synapsin-1	Total Synapsin-1	mAb	Rabbit	Cell Signaling (5297S)
Anti-Synaptotagmin	Total Synaptotagmin	mAb	Rabbit	Cell Signaling (14558S)
Anti-PSD93	Total PSD93	mAb	Rabbit	Cell Signaling (19046S)
Anti-PSD95	Total PSD95	mAb	Rabbit	Cell Signaling (3450S)
Anti-β-actin	Total β-actin	mAb	Mouse	Cell Signaling (3700S)
Anti-Rabbit 488	Donkey anti-Rabbit Alexa Fluor™ 488	pAb	Rabbit	Invitrogen (A21206)
Anti-Mouse 488	Donkey anti-Mouse Alexa Fluor™ 488	pAb	Mouse	Invitrogen (A21202)
HRP-Ribbit IgG	HRP Conjugated Goat Anti-Ribbit IgG	pAb	Rabbit	BOSTER (BA1050)
HRP-Mouse IgG	HRP Conjugated Goat Anti-mouse IgG	pAb	Mouse	BOSTER (BA1054)

## Data Availability

All data used in this study are available from the corresponding authors upon reasonable request.

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
