# Peer review of "Long-Term High-Fat Diet Consumption Induces Cognitive Decline Accompanied by Tau Hyper-Phosphorylation and Microglial Activation in Aging"

_nutrients, 2023, doi:10.3390/nu15010250_

Round 1

Reviewer 1 Report

The subject of the study is interesting and the scientific background has been described appropriately. I only suggest removing ”Pick’s disease” from the introduction, because that is now considered to be one variant of frontotemporal dementia.

The language is fluent and clear, although some minor typographic errors are found.

Abstract, results and discussion should be edited; There was no statistically significant cognitive function decline in fear conditioning or Barnes maze.

There is no discussion on the roles of different Tau phosphorylation sites. Which are the pathological states in which phosphorylation at p214, p396 or p422 is increased in mice? Which phosphorylation sites are modified by GSK-3β? Is its activation in hippocampus/cortex relevant with increased p214 in hippocampus and p396 & p422 in cortex?

The immunoblotting and immunofluorescence methodology sections do not give any info on the primary antibodies and insufficient info on the secondary antibodies. Manufacturers and product codes should be included, so that the validity of the results can be evaluated.

There is no description on the methodology of Sholl analysis. The results of the Sholl analysis do not seem to differ statistically although claimed so. There is only one significant point in the number of intersections at ca. 48 micrometres from the soma.

Author Response

1. The subject of the study is interesting and the scientific background has been described appropriately. I only suggest removing ”Pick’s disease” from the introduction, because that is now considered to be one variant of frontotemporal dementia.

Answer: Thank you for your advice. We have removed ”Pick’s disease” from the introduction (line 44).

2. The language is fluent and clear, although some minor typographic errors are found.

Abstract, results and discussion should be edited; There was no statistically significant cognitive function decline in fear conditioning or Barnes maze.

Answer: Thank you for pointing out our problems. We have revised the behavioral descriptions and edited the abstract (line 20), results (line 180-183) and discussion.

3. There is no discussion on the roles of different Tau phosphorylation sites. Which are the pathological states in which phosphorylation at p214, p396 or p422 is increased in mice? Which phosphorylation sites are modified by GSK-3β? Is its activation in hippocampus/cortex relevant with increased p214 in hippocampus and p396 & p422 in cortex?

Answer: We appreciate your insightful questions. Tau is hyperphosphorylated at over 40 serine/threonine sites, including S214, S396, S422, which generates 5-9 mol of phosphorous per mol of tau protein from the normal 2-3 mol. Hyperphosphorylation is considered to increase in the number of sites phosphorylation of the same tau molecule and in the number of tau molecules phosphorylated at the given sites. Therefore, it is currently difficult to identify which sites may be the most critical and initial pathological phosphorylation sites of tau in neurodegenerative diseases. Based on the kinase-recognized motifs, the serine/threonine kinases can be grouped into two classes: the PDPKs (proline-directed protein kinases) and the non-PDPKs. GSK-3β belong to the PDPKs. GSK-3β catalyses tau phosphorylation with high stoichiometry, including the sites of S214, S396, S422. Among multiple kinases, GSK-3β is the first identified tau kinase and it is the most strongly implicated in AD-like tau hyperphosphorylation at numerous sites. It has been included in the discussion (line 266-267).

4. The immunoblotting and immunofluorescence methodology sections do not give any info on the primary antibodies and insufficient info on the secondary antibodies. Manufacturers and product codes should be included, so that the validity of the results can be evaluated.

Answer: Thank you for your suggestion. The information about the primary and secondary antibodies are included in Table 1.

5. There is no description on the methodology of Sholl analysis. The results of the Sholl analysis do not seem to differ statistically although claimed so. There is only one significant point in the number of intersections at ca. 48 micrometers from the soma.

Answer: Thank you for your suggestion. We have supplemented the methodology of Sholl analysis in materials and methods (line143-145). There is one significant decrease in the number of intersections at ca. 48 micrometers from the soma and several non-signifiacant decrease in the number of intersections at ca. 10, 11,12, 13, 31,32, 33, 34  micrometers and so on.

Reviewer 2 Report

This work investigated the effect of a high fat diet on the cognitive decline in C57BL/6 mice, whilst assessing the levels of tau hyperphosphorylation, microglial activation and inflammatory response.  The authors concluded that there was a significant effect of consuming a HFD on cognitive decline linked with an increase in tau hyperphosphorylation, microglial activation and an increase in IL6 and TNFα.  Whilst the paper is well written and the results are well presented the data report is overstated or incorrect in several sections, which will influence the overall outcome and impact of the work.  These inaccuracies need to be rectified and discussed accordingly.

Specifically,

·         Abstract: line 20, states that fear conditioning and Barnes Maze, these outcomes were not significantly changed.

·         Abstract: line 21, only ps214 was significantly changed in the HC and ps422+ps396 in the cortex which should be correctly reported as such.

·         Abstract: line 25, only IL6 showed a significant change in the HC, TNFα did not, again an error in the reporting.

·         Figure 1. With exception to spatial recognition all of the other measures of cognition were insignificant.  Therefore, the title of the figure legend is inaccurate, and the supporting text overstates the outcome.

·         Results page 4: line 173, there is no significant difference, this needs to be rephrased and clearly articulated.

·         Line 180: the date does not support this conclusion.

·         Line 196: please be specific about what is being phosphorylated and its significance.

·         Line 199: the authors state that there is a significant difference in the images/figures 3A and B, there is no significant change based on the analysis.

·         Line 254, TNFα is not significantly changed.

Author Response

This work investigated the effect of a high fat diet on the cognitive decline in C57BL/6 mice, whilst assessing the levels of tau hyperphosphorylation, microglial activation and inflammatory response. The authors concluded that there was a significant effect of consuming a HFD on cognitive decline linked with an increase in tau hyperphosphorylation, microglial activation and an increase in IL6 and TNFα. Whilst the paper is well written and the results are well presented, the data report is overstated or incorrect in several sections, which will influence the overall outcome and impact of the work. These inaccuracies need to be rectified and discussed accordingly.

Specifically,

1. Abstract: line 20, states that fear conditioning and Barnes Maze, these outcomes were not significantly changed.

Answer: Thank you for pointing out our problems in abstract part. We have revised these descriptions in the abstract (line 20).

2. Abstract: line 21, only ps214 was significantly changed in the HC and ps422 + ps396 in the cortex which should be correctly reported as such.

Answer: Following this suggestion, We have revised the descriptions (line 21).

3. Abstract: line 25, only IL6 showed a significant change in the HC, TNFα did not, again an error in the reporting.

Answer: We appreciate your insightful questions. We have revised it (line 25).

4. Figure 1. With exception to spatial recognition all of the other measures of cognition were insignificant. Therefore, the title of the figure legend is inaccurate, and the supporting text overstates the outcome.

Answer: Thank you for your suggestion. We have modified the title of the figure legend (line 437) and revised the supporting text (line 180-182).

5. Results page 4: line 173, there is no significant difference, this needs to be rephrased and clearly articulated.

Answer: We appreciate your insightful questions. In the test phase of OPR task, HFD feeding animals showed a significantly reduced exploration of the displaced object. In the fear conditioning, the cued-induced freezing after 24 h was observed a tend to reduce in the HFD feeding mice compared to the ND. In a probe test of Barnes Maze, the HFD feeding group showed an increasing tendency of the primary errors. We have rephrased it (line 180-181).

6. Line 180: the date does not support this conclusion.

Answer: Thank you for your suggestion. We have modified it (line 180-182).

7. Line 196: please be specific about what is being phosphorylated and its significance.

Answer: We appreciate your insightful questions. Tau is hyperphosphorylated at over 40 serine/threonine sites, including S214, S396, S422, which generates 5-9 mol of phosphorous per mol of tau protein from the normal 2-3 mol. Hyperphosphorylation is considered to increase in the number of sites phosphorylation of the same tau molecule and in the number of tau molecules phosphorylated at the given sites. Therefore, it is currently difficult to identify which sites may be the most critical and initial pathological phosphorylation sites of tau in neurodegenerative diseases.

8. Line 199: the authors state that there is a significant difference in the images/figures 3A and B, there is no significant change based on the analysis.

Answer: Thank you for pointing out our problem. We have revised it (Line 194-195).

9. Line 254, TNFα is not significantly changed.

Answer: Thank you for your advice. We have revised it (Line 226).